# Promoting or Prohibiting? Investigating How Time Pressure Influences Innovative Behavior under Stress-Mindset Conditions

**DOI:** 10.3390/bs14020143

**Published:** 2024-02-17

**Authors:** Yufan Zhou, Jianwei Zhang, Wenfeng Zheng, Mengmeng Fu

**Affiliations:** 1Normal College, Qingdao University, Qingdao 266071, China; 2School of Humanities and Social Sciences, Beijing Institute of Technology, Beijing 100081, China; 3120205829@bit.edu.cn (W.Z.); mengmengfu@bit.edu.cn (M.F.)

**Keywords:** time pressure, stress mindset, innovative behavior, thriving at work

## Abstract

The existing empirical evidence on the relationship between time pressure and innovative behavior is paradoxical. An intriguing yet unresolved question is “When does time pressure promote or prohibit innovative behavior, and how?” We theorize that the paradoxical effect of time pressure on innovative behavior can be elucidated by the moderating role of stress mindset, and we also explore the mediating role of thriving at work. Our research involved a field study of 390 research and development personnel from eight enterprises and research institutes in China to test our proposed model. Results indicated that the stress-is-debilitating mindset negatively moderated the association between time pressure and thriving at work, while the stress-is-enhancing mindset positively moderated the link between time pressure and thriving at work. Furthermore, the findings demonstrated that the stress-is-debilitating mindset negatively moderated the indirect impact of time pressure on employees’ innovative behavior through thriving at work, while the stress-is-enhancing mindset positively moderated the indirect effect of time pressure on employees’ innovative behavior through thriving at work. The theoretical and practical implications of these findings are also discussed.

## 1. Introduction

Time pressure plays a crucial role in influencing employees’ innovative behavior [1,2]. However, there is limited consensus on the effects of time pressure on employees’ innovative behavior [3,4]. On the one hand, time pressure consumes cognitive resources and heightens reliance on familiar algorithms, potentially stifling innovative behavior [5]. On the other hand, time pressure may facilitate the collection of work-related knowledge and enhance task efficiency, thereby triggering innovative behavior in the pursuit of goals [6,7].

Although the relationship between time pressure and innovative behavior is complex, it remains unclear under what circumstances time pressure hinders or stimulates innovative behavior, and the mechanisms through which it does so are also unclear. The transactional model of stress and coping provides a theoretical insight, highlighting the tight link between stress outcomes and coping responses, which are influenced by the cognitive appraisal of environmental pressures [8]. Drawing upon the above model, time pressure may be appraised as a hindrance stressor, leading to a decrease in employees’ positive psychological states and further stifling innovative behaviors [9,10]. Additionally, time pressure can be perceived as a challenge stressor, encouraging employees to leverage valuable resources to creatively solve problems [7]. The inconsistent relationship between time pressure and innovative behavior underscores the importance of identifying moderators and intermediate mechanisms to understand when time pressure is appraised as challenging or threatening, and through which mechanisms time pressure impacts innovative behavior.

Following the transactional model of stress and coping, we have observed that the concept of stress mindset can offer a unique perspective in explaining the paradoxical relationship between innovative behavior and time pressure. The stress mindset can be understood as an individual’s beliefs regarding the debilitating or enhancing consequences of stress, and these beliefs can be categorized into two distinct mindsets: a stress-is-debilitating mindset and a stress-is-enhancing mindset [11]. Recent studies have shown that individuals’ context-specific stress appraisals are significantly influenced by their stress mindset [12]. Stress mindset may play the moderating role in the relationship between time pressure and innovative behavior. When employees with a stress-is-debilitating mindset perceive time pressure as a threat, they may experience reduced work engagement and exhibit less innovative behavior [13,14]. Conversely, employees with a stress-is-enhancing mindset tend to view time pressure as a challenge fostering grit and a focus on valuable information, which in turn contributes to creativity [7,15,16]. Therefore, exploring the moderating effects of stress mindsets can provide valuable insights into the intricate mechanisms that link time pressure to innovative behavior.

Furthermore, the transactional model of stress and coping also emphasizes that the stress coping process is initiated following cognitive appraisals, and the ultimate stress outcomes are influenced by one’s coping strategies [15]. According to this theoretical model, we argue that coping strategy plays a crucial mediating role in how time pressure impacts innovative behavior. One coping strategy closely linked to employees’ stress appraisals could be thriving at work [17]. Thriving at work reflects a positive psychological state in the workplace, which has been shown to benefit employees’ innovative behavior [10,18]. While previous research has frequently linked time pressure to employees’ negative psychological states, it is worth noting that time pressure can also stimulate positive working states, such as knowledge scanning and job crafting, which are associated with thriving and work and contribute to innovative behavior [4,7,19]. In this regard, we emphasize that thriving at work is influenced by the interactive effects of time pressure and stress mindsets, and in turn, has an impact on innovative behavior.

Specifically, individuals with a strong stress-is-debilitating mindset may perceive time pressure as leading to negative outcomes and resource depletion [20]. They may focus on conserving resources and may not be as energetic or diligent, which hinders their ability to thrive at work and subsequently reduces innovative behavior [21]. On the other hand, when individuals perceive that time pressure results in positive outcomes and resource gains, they may exhibit vibrancy and conscientiousness. Consequently, employees with a stress-is-enhancing mindset are more likely to acquire additional knowledge and invest more energy in their work to obtain more resources, which contributes to innovative behavior [11,22].

As a result, we propose that the contradictory relationship between time pressure and innovative behavior can be mediated by thriving at work, and the complexity of this relationship is elucidated by the impact of stress mindsets, which moderate the mediating mechanism.

## 2. Literature and Hypothesis Development

### 2.1. Time Pressure and Innovative Behavior

The term employee innovative behavior refers to employees’ deliberate introduction or application of novel and valuable ideas, products, processes, and procedures within their work [23,24]. One of the most significant antecedent factors influencing innovative behavior is time pressure [1,2]. Time pressure, defined as the perceived stress associated with the necessity to complete tasks within a specified time limit [25], has been identified in previous research to have both promoting and inhibiting effects on innovative behavior [1,26]. However, the reasons underlying this contradictory relationship require further investigation. Recently, researchers have shown increased interest in unraveling the underlying mechanisms of this contradictory relationship between time pressure and innovative behavior [3,4].

The present research focuses on the dual mechanism through which time pressure influences innovative behavior. The transactional model of stress and coping suggests that stress outcomes are influenced by an individual’s cognitive appraisal of stress [8,27]. In accordance with this model, we posit that an individual’s stress mindset plays a pivotal role in understanding the impact of time pressure on innovative behavior. While some research has suggested that time pressure may not be conducive to innovative behavior [5,28,29], it is overly simplistic to assume that the relationship between time pressure and innovative behavior is universally negative.

Under time pressure, individuals may have varying appraisals of resource loss or gain due to their different stress mindsets [30]. Specifically, individuals with a stress-is-debilitating mindset believe that stress leads to debilitating consequences, and they tend to focus on negative information that reinforces their negative beliefs [11]. This belief, in turn, leads them to reduce self-resource input and protect against resource loss [13]. Employee innovative behavior is significantly dependent on their engagement with work resources, including cognitive, emotional, and physical energy [14]. Therefore, individuals with a stress-is-debilitating mindset are more likely to maintain or even decrease their concentration of work resources under time pressure, which is not conducive to innovative behavior.

Conversely, individuals with a stress-is-enhancing mindset believe that experiencing pressure will ultimately result in enhancing outcomes. In this perspective, they may view time pressure as a resource-enhancing stressor, motivating them to adopt effective coping strategies and engage their work resources to obtain more assistance. In this context, even when employees perceive high levels of time pressure, they are more inclined to invest additional work resources, leading to improved work performance and a sense of physiological thriving, which ultimately results in more innovative behaviors [31,32].

In summary, time pressure does not necessarily impede employees’ innovative behavior. Instead, the relationship between time pressure and innovative behavior is contingent on stress mindsets, which can lead individuals to appraise and cope with time pressure differently. In the subsequent sections, we focus on thriving at work as a coping strategy that reflects stress appraisal and responses. Thriving at work serves as a mediator that explains the impact of time pressure on innovative behavior [32]. We examine stress mindsets, specifically the stress-is-debilitating mindset and the stress-is-enhancing mindset, as moderators in the relationship between time pressure and innovative behavior via the pathway of thriving at work.

### 2.2. Time Pressure and Thriving at Work

Thriving at work is a positive psychological state characterized by the concurrent experience of “vitality” and “learning” [10]. According to a socially embedded model, contextual factors, such as pressure and organizational climate, along with personal resources, such as cognitive appraisal, are crucial antecedents of thriving at work [10,17].

The conservation of resources theory posits that when individuals’ resources are depleted, they enter a defensive mode to protect themselves, often leading to defensive or burnout responses [20,21]. In this context, time pressure is likely to be negatively associated with thriving at work. Time is considered a vital work resource for employees. However, as time pressure increases, individuals gradually deplete their time resources and become fatigued. In such situations, to compensate for the resource loss, employees tend to invest fewer work resources and struggle to thrive at work. Additionally, meeting deadlines, a common work requirement, may compel them to expend their resources to complete overloaded tasks [33]. As cognitive and emotional resources are continually drained, employees may exhibit negative behaviors, such as burnout, to safeguard their remaining resources [34]. Simultaneously, they must invest personal time and energy to adjust their negative behavioral disposition, which is typically disapproved by organizations [35], making it challenging to achieve a high level of thriving at work.

Although the conservation of resources theory suggests a likely negative relationship between time pressure and thriving at work, it also indicates that employees’ working state is influenced by human energy [36]. Recent research on thriving at work further supports this viewpoint [37]. In other words, the negative association between time pressure and thriving at work does not provide a complete understanding of this relationship. Building on previous studies, the following section theorizes that time pressure can either promote or hinder thriving at work, contingent upon employees’ stress mindsets, which shape their perceptions of stress as debilitating or enhancing. Specifically, we will focus on the moderating role of the stress-is-debilitating mindset and the stress-is-enhancing mindset.

### 2.3. Interactive Effects of Time Pressure and Stress Mindset

Stress mindset plays a significant role in shaping individual’s response to stress [38,39]. Previous research has shown that the physical and psychological outcomes resulting from an objective level of stress are often outweighed by individuals’ perceptions that their responses to stress are driven by their appraisal of its effects [11]. For example, employees with a stress-is-debilitating mindset typically fear stress and challenges because they believe stress may have adverse effects, leading them to adopt passive coping strategies [40]. Conversely, employees with a stress-is-enhancing mindset are generally unafraid of stress and challenges because they believe stress can facilitate personal development and foster positive behaviors. Therefore, we propose that time pressure and a stress mindset interact to affect thriving at work.

Specifically, we predict an interactive effect of time pressure and the stress-is-debilitating mindset on thriving at work. A stronger stress-is-debilitating mindset is likely to exacerbate the detrimental impacts of time pressure on thriving at work. Employees with a stress-is-debilitating mindset believe that stress has destructive effects and tend to experience anxiety or irritability. They often resort to emotional expression and emotional support when they perceive pressure [41]. These stress responses can deplete their work resources, which is not conducive to thriving or problem-solving.

Furthermore, a stress-is-debilitating mindset prompts employees to focus on negative information in their work context. This focus on negative information reinforces their negative beliefs about stress, leading them to adopt passive or avoidant coping strategies and exhibit stress-avoidant emotions and behaviors [42]. Consequently, a stress-is-debilitating mindset directs employees to engage in fewer proactive learning behaviors. Additionally, employees with this mindset often lack autonomous motivation and find it challenging to identify opportunities for learning and self-development. Their limited enthusiasm for acquiring new knowledge and skills makes it difficult for them to experience vitality [43].

In contrast, employees with a less strong stress-is-debilitating mindset are likely to manifest less harmful cognitive, emotional, and behavioral responses to time pressure. Consequently, time pressure would lead to fewer stress-avoidant emotions and behaviors. Therefore, we anticipate that time pressure may have a negative impact on thriving at work when employees possess a stronger stress-is-debilitating mindset. Our expectations are as follows:

**Hypothesis** **1.**
*A stress-is-debilitating mindset moderates the relationship between time pressure and thriving at work, such that this negative relationship is stronger when the employees have a stronger stress-is-debilitating mindset.*


We also propose that the stress-is-enhancing mindset can play a moderating role in the relationship between time pressure and thriving at work. A stress-is-enhancing mindset directs individuals to hold positive beliefs about time pressure. This cognitive appraisal makes employees more optimistic about time pressure and less prone to experiencing anxiety, depression, and other negative emotions when they are under pressure. It enables employees to work with enthusiasm and energy [39]. People with a stress-is-enhancing mindset typically exhibit commendable characteristics, such as grit, optimism, and mindfulness [11]. These traits empower employees to persistently seek solutions to problems, even under pressure. Problem solving is often accompanied by knowledge acquisition and skill improvement, which contribute to thriving at work [39].

This type of stress mindset can also stimulate autonomous motivation and a learning-goal orientation, thereby increasing employees’ work engagement and enhancing their vitality and learning [44]. Furthermore, employees with a stronger stress-is-enhancing mindset are more likely to seek performance feedback from their supervisors or colleagues. These feedback mechanisms help them adjust their behaviors and psychological state and learn how to tackle challenges [43,45]. Consequently, employees with a stronger stress-is-enhancing mindset are more likely to thrive at work. In contrast, individuals with a weaker stress-is-enhancing mindset may struggle to cultivate positive work emotions, motivation, and behaviors and are less likely to experience thriving at work. Overall, we hypothesize that:

**Hypothesis** **2.**
*A stress-is-enhancing mindset moderates the relationship between time pressure and thriving at work, such that this positive relationship is stronger when employees have a stronger stress-is-enhancing mindset.*


### 2.4. Time Pressure, Thriving at Work, and Innovative Behavior

The interactive effects of time pressure and stress mindsets can be further channeled into innovative behavior through the mediating role of thriving at work. Previous studies have highlighted that thriving at work is a crucial precursor to innovative behavior [30,46]. Individuals who are thriving at work typically exude vitality and experience more positive emotions, which are conducive to creative thinking and innovative problem-solving [47]. Additionally, positive emotions often lead individuals to build their social and psychological resources, ultimately enhancing their work engagement and fostering innovative behaviors [48].

Moreover, vitality is closely linked to autonomous motivation, a critical precursor to individual innovative behaviors [30]. Thriving at work motivates employees to acquire more knowledge and skills, which are invaluable for bringing creative ideas to life. Learning not only stimulates employees to generate creative ideas, experiment with new approaches, and enhance existing work practices, but also helps them acquire professional knowledge and refine their working skills. The enhancement of employees’ knowledge and skills plays a significant role in realizing creative methods [22].

Furthermore, thriving at work fosters a positive response to work challenges. Employees become adept at identifying and addressing current work problems with flexibility and adaptability. This ultimately leads to an increase in their innovative behaviors [30].

Integrating the reasoning presented above, we predict that time pressure has a mediated relationship with innovative behavior through thriving at work. The nature of this mediated relationship differs under the two types of stress mindsets. When time pressure is perceived as a more obstructive factor, employees may develop a negative perception of the stressful situation, leading to anxiety or depression. Consequently, they may choose passive coping strategies and deplete their work resources to manage negative emotions and motivations. These adverse reactions are believed to contribute to the negative association between time pressure and thriving at work. Furthermore, low levels of thriving at work are not conducive to creative problem-solving, thereby inhibiting employees’ innovative behavior. Therefore, in the case of employees with a stronger stress-is-debilitating mindset, time pressure is negatively associated with innovative behavior through its impact on their thriving at work. Thus, we propose the following moderated mediation model:

**Hypothesis** **3.**
*A stress-is-debilitating mindset moderates the indirect relationship between time pressure and innovative behavior via thriving at work, such that this negative relationship is stronger when the employees have a stronger stress-is-debilitating mindset.*


When employees have a stronger stress-is-enhancing mindset, they tend to hold positive views and respond positively to time pressure. They believe that stressful events can enhance their performance, health, and well-being, leading them to choose active coping strategies to solve work-related problems. Consequently, they exhibit a high level of vitality and learning. This improvement in thriving at work is conducive to creative thinking and autonomous motivation, ultimately enhancing employees’ innovative behavior. Therefore, in the case of employees with a stronger stress-is-enhancing mindset, time pressure is positively associated with innovative behavior through its impact on their thriving at work. Hence, we expect the following moderated mediation model:

**Hypothesis** **4.**
*A stress-is-enhancing mindset moderates the indirect relationship between time pressure and innovative behavior via thriving at work, such that this positive relationship is stronger when the employees have a stronger stress-is-enhancing mindset.*


Our theoretical model is depicted in Figure 1.

## 3. Materials and Methods

### 3.1. Sample and Procedures

To test our hypothesis, the convenience sampling method was used to select participants involved in research and development activities across eight Chinese enterprises and research institutes. We specifically chose these settings because employees in these environments are frequently tasked with engaging in innovative behaviors and commonly experience time pressure. A total of two rounds of data were collected through anonymous paper-and-pencil questionnaires, with an 8-week interval between the two rounds. In Round 1, we gathered demographic variables, the independent variable of time pressure, and the moderating variables of stress mindsets. In Round 2, we evaluated the variables of thriving at work and innovative behavior. To match the responses across the two rounds, participants were requested to record the last six digits of their phone numbers. The final matched sample comprised 390 employees (response rate: 86.67%).

Of the 390 final respondents, 31.0% were aged 20–30, 42.8% were aged 31–40, 19.0% were aged 41–50, and 7.2% were aged over 51. The gender distribution included 41.3% females and 58.7% males. Organizational tenure was distributed as follows: 31.3% had less than 5 years of tenure, 24.1% had 6 to 10 years, 18.5% had 11 to 15 years, 8.7% had 16 to 20 years, and 17.4% had more than 20 years of tenure. Education levels consisted of 6.7% with a technical college degree, 59.7% with a bachelor’s degree, and 33.6% with a master’s degree.

### 3.2. Measures

In our study, measurements for the variables were selected from established scales published in authoritative journals. These scales were translated and revised by researchers in management and psychology, tailored to the context of Chinese research and development enterprises and institutes. Before the formal questionnaire study and survey administration, small-scale tests were conducted. After confirming the reliability and validity of all scales, the final questionnaire was developed. Except where noted otherwise, this study used five-point Likert-type scales ranging from 1 (completely disagree) to 5 (completely agree).

Time Pressure. Employees’ perceived time pressure was assessed using the three-item scale proposed by Durham et al. (2000) [49], ranging from 1 (totally disagree) to 7 (totally agree). A sample item is “I felt that I was working under excessive time pressure”. The Cronbach’s α was 0.706.

Stress Mindset. We measured the stress-is-debilitating mindset and stress-is-enhancing mindset using the scales developed by Crum et al. (2013) [11]. Respondents were asked to indicate the degree to which the stress-is-debilitating mindset and stress-is-enhancing mindset items related to their personal views. Four items were used to measure the stress-is-debilitating mindset (e.g., “Experiencing stress depletes my health and vitality”). Four items were used to measure the stress-is-enhancing mindset (e.g., “Experiencing stress enhances my performance and productivity”). The estimated Cronbach’s α values of the stress-is-debilitating and the stress-is-enhancing mindset were 0.736 and 0.788, respectively.

Thriving at Work. To measure employees’ thriving at work, we used the 10-item scale developed by Porath et al. (2012) [50]. Items included “I continue to learn more and more as time goes by” and “I have energy and spirit.” The Cronbach’s α of this scale was 0.829.

Innovative Behavior. Employees’ innovative behavior was assessed using Scott and Bruce’s (1994) six-item scale [23]. Respondents reacted to items such as “Generates creative ideas”. The Cronbach’s α was 0.903.

Control Variables. Given that previous studies have suggested that demographic variables may confound the hypothesized relations, we controlled for demographic variables of age, gender, organizational tenure, and educational level because these factors have the propensity to influence thriving at work and innovative behavior [51].

### 3.3. Analysis Strategy

Our study used SPSS 25.0 and AMOS 24.0 software for data analysis. First, we assessed the reliability and descriptive statistics of the variables using SPSS. Second, exploratory factor analysis and confirmatory factor analysis were employed to examine common method deviation with the assistance of both SPSS and AMOS. Finally, hierarchical regression analysis and bootstrap analysis were conducted using SPSS and PROCESS procedures to examine the hypothesized relationships [52].

## 4. Results

### 4.1. Common Method Deviation Test

To ensure that common method variance (CMV) was not a concern in our study, we took measures in both the measurement design and statistical analysis. First, we employed a two-round data collection with an 8-week interval to reduce the impacts of CMV [53]. Second, we utilized anonymous questionnaires, incorporated mutually exclusive questions, and randomized the arrangement of questions. Third, we employed Harman’s single-factor method by subjecting all items to exploratory factor analysis, and we examined the unrotated factor solution. The results indicated that the maximum characteristic root-explanation variance was 25.24%, which did not account for more than half of the total variance of 61.856%. Subsequently, we equated the baseline model with and without a latent CMV. The results showed that the differences between these models were not statistically significant (ΔCFI = 0.006, ΔTLI = 0, ΔRMSEA = 0). Consequently, both the results of Harman’s single-factor analysis and the assessment of the baseline model against alternative models through confirmatory factor analysis (the results shown in Table 1) indicated that CMV did not pose a substantial threat to our study.

### 4.2. Descriptive Analysis

Table 2 presents the means, standard deviations, and correlation coefficients for the key variables. The results indicate that time pressure has a negative correlation with thriving at work (*r* = −0.116, *p* < 0.01). In contrast, thriving at work exhibits a positive correlation with innovative behavior (*r* = 0.406, *p* < 0.01). Additionally, the data reveal that the stress-is-debilitating mindset is inversely correlated with both thriving at work (*r* = −0.261, *p* < 0.01) and innovative behavior (*r* = −0.180, *p* < 0.01). Conversely, the stress-is-enhancing mindset shows a positive correlation with thriving at work (*r* = 0.263, *p* < 0.01) and innovative behavior (*r* = 0.441, *p* < 0.01).

### 4.3. Hypothesis Test

To mitigate issues related to multicollinearity, all variables in the regression equation were standardized. Control variables, such as age, gender, organizational tenure, and education, were included in the analysis. The assessment of the hypothesized relationships involved constructing a moderated mediation model, which was subsequently analyzed. Following the guidelines of Edwards and Lambert (2007) [54], we initially assessed the moderating impact of the stress-is-debilitating mindset and the stress-is-enhancing mindset on the relationship between time pressure and thriving at work. Following this, we proceeded to assess the moderated mediation model concerning the relationship between time pressure and innovative behavior. Table 3 presents the regression results for the hypothesized model.

Hypothesis 1 posited that the stress-is-debilitating mindset moderates the relationship between time pressure and thriving at work, with a stronger negative relationship when employees have a stronger stress-is-debilitating mindset. Models 2 and 4 revealed that the interaction term involving time pressure and the stress-is-debilitating mindset was indeed negatively associated with thriving at work (β = −0.110, *p* < 0.05), supporting Hypothesis 1.

To further explore the moderating effect of time pressure and the stress-is-debilitating mindset on thriving at work, we conducted a simple slope test. As depicted in Figure 2, the stress-is-debilitating mindset moderated the effects of time pressure on thriving at work. When employees possessed a stronger stress-is-debilitating mindset (+1 SD), the relationship between time pressure and thriving at work was significantly negative (simple slope = −0.108, *t* = −3.061, *p* < 0.01). Conversely, when they had a less strong stress-is-debilitating mindset (−1 SD), the association between time pressure and thriving at work became nonsignificant (simple slope = −0.015, *t* = −0.465, *p* > 0.05).

Hypothesis 2 posits that the stress-is-enhancing mindset moderates the relationship between time pressure and thriving at work, such that this positive relationship is stronger when employees have a stronger stress-is-enhancing mindset. Models 2 and 6 provided support for this hypothesis, as the interaction term involving time pressure and the stress-is-enhancing mindset was significantly positively related to thriving at work (β = 0.318, *p* < 0.001).

To delve further into the moderating impact of time pressure and the stress-is-enhancing mindset on thriving at work, a simple slope test was conducted. As illustrated in Figure 3, the stress-is-enhancing mindset moderated the effects of time pressure on thriving at work. When employees possessed a stronger stress-is-enhancing mindset (+1 SD), the positive relationship between time pressure and thriving at work was significant (simple slope = 0.074, *t* = 2.332, *p* < 0.05). Conversely, when they exhibited lower levels of the stress-is-enhancing mindset (−1 SD), the relationship between time pressure and thriving at work became significantly negative (simple slope = −0.189, *t* = −5.663, *p* < 0.001).

To examine the conditional indirect effects, we utilized the Monte Carlo simulation approach to calculate the 95% confidence interval (CI). Table 4 provides an overview of the model results.

Hypothesis 3 postulates that the stress-is-debilitating mindset moderates the indirect relationship between time pressure and innovative behavior through thriving at work, such that this negative relationship is stronger when employees have a stronger stress-is-debilitating mindset. As displayed in Table 4, the index of moderated mediation is significant (index = −0.043, 95% CI [−0.086, −0.004]), lending strong support to Hypothesis 3. Furthermore, the results reveal that when the stress-is-debilitating mindset was stronger (+1 SD), the indirect effect of time pressure on innovative behavior was significantly negative (index = −0.085, 95% CI [−0.146, −0.035]). Conversely, when the stress-is-debilitating mindset was weaker (−1 SD), the indirect effect of time pressure on innovative behavior was not significant (index = −0.012, 95% CI [−0.063, 0.039]).

Hypothesis 4 further predicts that the stress-is-enhancing mindset moderates the indirect relationship between time pressure and innovative behavior through thriving at work, resulting in a stronger positive relationship when employees have a stronger stress-is-enhancing mindset. As presented in Table 4, the moderated mediation index is significant (index = 0.118, 95% CI [0.078, 0.173]). When the stress-is-enhancing mindset was stronger (+1SD), the indirect effect of time pressure on innovative behavior was significantly positive (index = 0.058, 95% CI [0.010, 0.118]). Conversely, when the stress-is-enhancing mindset was weaker (−1 SD), the indirect effect of time pressure on innovative behavior was significantly negative (index = −0.149, 95% CI [−0.217, −0.094]). The results provide robust support for Hypothesis 4.

## 5. Discussion

This study aimed to determine under what circumstances time pressure either hinders or fosters innovative behavior. We explored the moderating impact of stress mindsets and the mediating role of thriving at work. The results indicated that the stress-is-debilitating mindset amplified the adverse direct association between time pressure and thriving at work, as well as accentuated the negative indirect impact of time pressure on innovative behavior through thriving at work. Conversely, the stress-is-enhancing mindset counteracted the aforementioned negative relationships. When employees held a stronger stress-is-enhancing mindset, time pressure exhibited a positive correlation with thriving at work and exerted a positive indirect influence on innovative behavior through thriving at work.

### 5.1. Theoretical Implications

The current study has several implications for theory. First, our study makes a significant contribution to the existing literature on time pressure and its relationship with innovative behavior. This relationship has been previously examined with mixed findings—some studies suggest that time pressure inhibits employees’ innovative behavior, while others propose that it can have motivational effects on creativity [7,29,55]. Despite this body of research highlighting time pressure as a double-edged sword for innovative behavior, there remains a lack of clarity regarding why time pressure exerts paradoxical effects on employees’ creative performance [3,4]. In our study, we aim to unravel this paradox by introducing stress mindsets as pivotal factors that can modify the impact of time pressure.

In line with the transactional model of stress and coping as well as implicit theories [8], stress mindsets offer a framework for appraising the outcomes associated with time pressure, subsequently transforming its influences. Our research revealed that time pressure has a negative influence on innovative behavior in individuals with a strong stress-is-debilitating mindset. Importantly, when employees possess a less strong stress-is-debilitating mindset, the impact of time pressure on innovative behavior is not significant. Conversely, for employees with a stronger stress-is-enhancing mindset, time pressure is more likely to stimulate innovative behavior. Therefore, our study expands the existing body of research on time pressure by underscoring the crucial moderating role of stress mindsets in the relationship between time pressure and innovative behavior.

Second, our study contributes to the growing body of research on stress mindset by illustrating that stress mindset plays a distinct and influential role in the relationship between time pressure and innovative behavior. Prior research has acknowledged that pressure can elicit either challenging or threatening effects on individuals’ psychological well-being and performance [56]. It has aimed to uncover how stressors lead to various outcomes [3,8]. However, a significant portion of stress intervention studies has primarily focused on cognitive appraisals, motivation, and coping [41,57], often overlooking the pivotal role of stress mindsets. According to the stress mindset theory, stress mindset significantly influences how individuals approach and experience stress, as emphasized by Crum et al. [11]. The research on stress mindset calls for further investigation into the effects of stress mindsets under varying types of pressure [12]. Our research responds to the call for exploring the effects of stress mindsets and provides an explanation for the inconsistent impact of time pressure on innovative behavior.

Third, our study contributes to the literature on thriving at work by highlighting its mediating role in the relationship between time pressure and innovative behavior. Specifically, this study delves into how time pressure has paradoxical effects on thriving at work and seeks to elucidate its collaborative impact on employees’ innovative behavior. This contribution is aligned with the call for research aimed at understanding how thriving at work, as a mediator, relates to work resources and innovative behavior [30].

Moreover, our study underscores the significance of stress mindsets as crucial individual resources for thriving at work. Thriving at work is commonly regarded as a resource-oriented psychological state [58]. While prior research has predominantly focused on the influence of contextual resources, such as organizational culture and leadership behaviors [46,57,59], it has often overlooked the potent impact of employees’ personal resources. This study expands the antecedent research on thriving at work by demonstrating that employees’ ability to thrive at work, particularly under high levels of time pressure, hinges on their stress-is-debilitating and stress-is-enhancing mindsets.

### 5.2. Practical Implications

Below, we highlight multiple practical implications. First, the findings of our study offer important insights into the management of employees’ stress mindsets. Notably, time pressure can either hinder or foster thriving at work and innovative behavior, contingent upon distinct stress-mindset influences. For instance, employees facing high levels of time pressure often grapple with a stress-is-debilitating mindset, which accentuates their focus on the adverse aspects of pressure. Therefore, managers should pay close attention to employees’ stress mindsets. Identifying employees’ stress mindset types based on their work behaviors—such as proactive feedback seeking and continuous learning, characteristic of the stress-is-enhancing mindset—is essential [11]. Managers can then tailor their approach, assigning more challenging tasks to employees with the stress-is-enhancing mindset and matching those with the stress-is-debilitating mindset to practical responsibilities. Furthermore, managers can contribute to the development of employees’ stress-is-enhancing mindset by emphasizing the importance of learning and self-development, encouraging a focus on career growth rather than just performance outcomes.

Second, our research provides valuable suggestions for time pressure management within organizations, particularly in high levels of time pressure work contexts. Employees with a stronger stress-is-debilitating mindset may tend to adopt avoidance coping strategies, such as engaging in non-work-related activities and dwelling on negative contextual information [12]. In contrast, employees with a stronger stress-is-enhancing mindset are more likely to employ problem-solving coping strategies, concentrating on the positive facets of stress. Additionally, our study found that when employees have a less strong stress-is-debilitating mindset, time pressure does not impede their thriving at work and innovative behavior. Given these individual differences in coping with time pressure, managers aiming to effectively motivate their teams should consider these varying responses. For employees with a stronger stress-is-enhancing mindset or a less strong stress-is-debilitating mindset, managers can assign urgent tasks and provide prompt feedback to stimulate their positive emotions and behaviors. Conversely, for employees with a strong stress-is-debilitating mindset, managers should calibrate task difficulty and allocate assignments that align with employees’ competencies. This approach can boost self-efficacy and encourage proactive behavior.

Lastly, our research underscores the critical role of thriving at work in fostering innovative behavior among employees. Managers must take a scientific approach to promote thriving at work, aligning with the conservation of resources theory, which highlights the significance of adequate work resources in stimulating employees’ thriving [60]. To provide career development resources, managers can offer employees more opportunities, such as professional skills training, leadership development programs, and access to a valuable career development platform. Moreover, establishing a healthy organizational climate that encourages interpersonal resources is vital. Organizing cultural and family activities, enhancing employee relationships, and fostering an atmosphere of open communication and mutual trust are all integral in this regard.

### 5.3. Limitations and Future Directions

Despite the contributions and insights offered by this research, it is essential to acknowledge certain limitations that should be considered when interpreting the findings. First, the cross-sectional design of data collection limits our ability to establish causal relationships between variables. While the results support our hypotheses, the cross-sectional nature of the study limits our ability to establish causal relationships between variables. The reliance on self-reported measures for our study’s focal variables also raises concerns about CMV, a potential source of bias [61]. Future studies could enhance the robustness of our findings through more rigorous research designs. For instance, researchers may opt for a longitudinal design, enabling the assessment of changes over time and providing stronger evidence of causality. Additionally, the use of objective data to measure employees’ innovative performance would mitigate the issues associated with self-reported measures.

Moreover, it is crucial to recognize the potential limitations associated with the sample and context of our study. The participants of our study were Chinese research and development employees. While innovative behaviors are notably prevalent in the research and development sector, it is essential to acknowledge that these behaviors are not confined to a single industry. Therefore, the generalizability of our findings beyond this specific context may be constrained. To address this limitation and enhance the external validity of our research, we recommend that future studies explore the hypothesized model in diverse populations and settings. Collecting data from a wide range of groups and industries will serve to mitigate generalizability concerns and broaden the applicability of our study findings. By expanding the scope of research to include other demographic groups and sectors, researchers can make more comprehensive contributions to both theory and practice in the field.

Finally, our research explored the moderating impact of stress mindsets on the relationship between time pressure and innovative behavior. However, from the practical perspective of assisting employees in coping with stress, mindfulness is also a worthwhile factor to explore as a potential moderator in the relationship between time pressure and innovative behavior [62]. We focused solely on the moderating influence of personal factors and did not examine potential moderating effects related to contextual and leadership factors, such as organizational climate and leader–member exchange [60,63]. Thus, further research on the contextual and leadership moderating mechanisms is warranted. Additionally, varying levels of time pressure may result in different stress outcomes. Future studies could investigate the interactive effects of these diverse time pressure levels and individual personality characteristics on innovative behavior, drawing insights from arousal theory [64].

## 6. Conclusions

Our study enriches the literature on time pressure and its impact on innovative behavior by introducing and validating a model that highlights the differential effects of time pressure based on two distinct stress mindsets: the stress-is-debilitating mindset and the stress-is-enhancing mindset. Furthermore, we put forth a moderated mediation model, revealing that the direction of the mediated relationship between time pressure, thriving at work, and innovative behavior varies depending on the individual’s stress mindset. Our findings provide valuable insights for managers and practitioners aiming to leverage time pressure as a means to enhance employees’ workplace learning, vitality, and innovative behaviors. To this end, it is recommended that organizations consider not only the level of time pressure but also interventions to foster a stress-is-enhancing mindset among employees.

## Figures and Tables

**Figure 1 behavsci-14-00143-f001:**
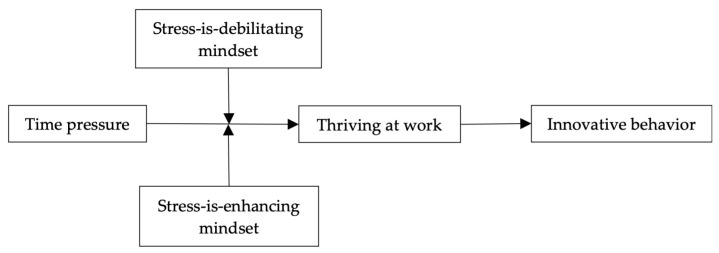
Theoretical model.

**Figure 2 behavsci-14-00143-f002:**
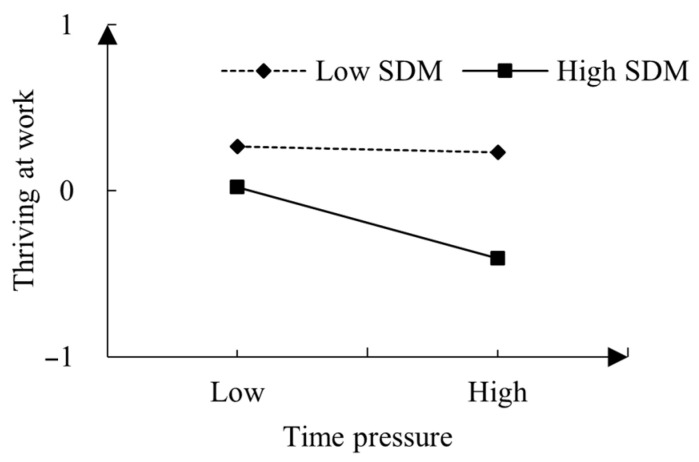
Interaction between time pressure and SDM (stress-is-debilitating mindset) on thriving at work.

**Figure 3 behavsci-14-00143-f003:**
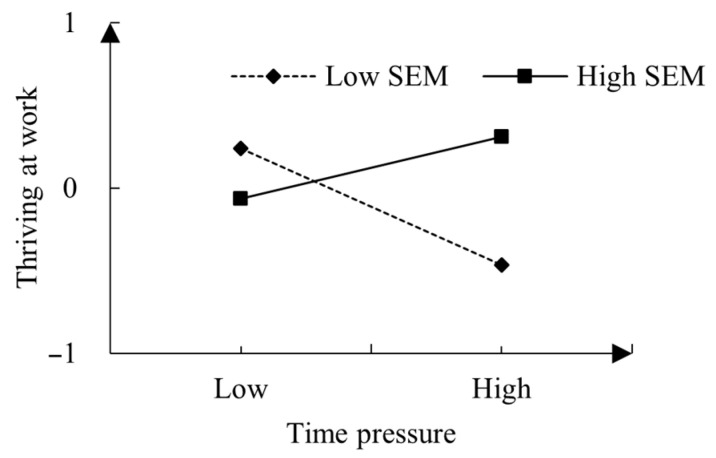
Interaction between time pressure and SEM (stress-is-enhancing mindset) on thriving at work.

**Table 1 behavsci-14-00143-t001:** Comparison of measurement models.

Model	*χ* ^2^	df	*χ*^2^/df	RMSEA	GFI	CFI	TLI
Baseline model	934.322	314	2.976	0.071	0.920	0.923	0.906
Four-factor model ^1^	1131.749	319	3.548	0.081	0.880	0.861	0.839
Four-factor model ^2^	1113.244	319	3.490	0.080	0.879	0.866	0.846
Three-factor model	1494.517	323	4.627	0.097	0.765	0.718	0.693
Two-factor model	1873.817	326	5.748	0.110	0.713	0.627	0.598
One-factor model	2570.940	328	7.838	0.133	0.596	0.459	0.421

Note: Baseline model = five-factor model; ^1^ four-factor model = combine time pressure and stress-is-debilitating mindset; ^2^ four-factor model = combine time pressure and stress-is-enhancing mindset; three-factor model = combine time pressure, stress-is-debilitating mindset and stress-is-enhancing mindset; two-factor model = combine time pressure, stress-is-debilitating mindset, stress-is-enhancing mindset and thriving at work; one-factor model = combine five variables. Abbreviations: RMSEA = root mean square error of approximation; GFI = goodness-of-fit index; CFI = comparative fit index; TLI = Tucker–Lewis Index.

**Table 2 behavsci-14-00143-t002:** Descriptive statistics and correlation matrix.

Variable	Mean	SD	1	2	3	4	5	6	7	8	9
1. Age	2.023	0.888									
2. Gender	1.413	0.493	−0.004								
3. OT	3.454	1.600	−0.853 **	0.036							
4. Education	2.287	0.612	−0.291 **	−0.036	0.375 **						
5. TP	4.689	0.832	0.057	−0.100 *	0.052	−0.039	(0.755)				
6. SDM	3.444	0.686	0.057	0.052	0.054	−0.113 *	0.003	(0.748)			
7. SEM	3.462	0.712	0.035	−0.139 **	0.008	−0.074	−0.038	−0.003	(0.781)		
8. TAW	3.349	0.478	−0.139 **	−0.019	−0.148 **	0.032	−0.116 **	−0.261 **	0.263 **	(0.735)	
9. IB	3.443	0.619	−0.024	−0.052	0.042	0.058	−0.015	−0.180 **	0.441 **	0.406 **	(0.819)

Note: 1 = male and 2 = female. The square root of AVE for each variable is shown in parentheses; N = 390, * *p* < 0.05, ** *p* < 0.01. Abbreviations: OT, organizational tenure; TP, time pressure; SDM, stress-is-debilitating mindset; SEM, stress-is-enhancing mindset; TAW, thriving at work; IB, innovative behavior; SD, standard deviation.

**Table 3 behavsci-14-00143-t003:** Regression analysis results.

Variables	Thriving at Work					
	Model 1	Model 2	Model 3	Model 4	Model 5	Model 6
Control variables						
Age	−0.026	−0.019	−0.047	−0.044	0.004	−0.003
Gender	−0.016	−0.027	−0.014	−0.005	0.011	−0.007
Organizational tenure	−0.118	−0.112	−0.122	−0.121	−0.086	−0.027
Education	−0.046	−0.043	−0.028	−0.022	−0.068	−0.120
Independent variable						
Time pressure		−0.110 *	−0.107 *	−0.126 *	−0.097 *	−0.088
SDM			−0.257 ***	−0.221 ***		
SEM					0.264 ***	0.120 *
Interaction						
Time pressure × SDM				−0.110 *		
Time pressure × SEM						0.318 ***
R^2^	0.023	0.035	0.100	0.111	0.103	0.182
ΔR^2^	0.023	0.012	0.065	0.076	0.068	0.147
F	2.308	2.808 *	7.106 ***	6.780 ***	7.314 ***	12.182 ***

Note: N = 390, * *p* < 0.05, *** *p* < 0.001. Abbreviations: SDM, stress-is-debilitating mindset; SEM, stress-is-enhancing mindset.

**Table 4 behavsci-14-00143-t004:** Conditional indirect effect.

	Effect	SE	95% CI	
			Lower	Upper
SDM as a moderator				
Index of moderated mediation	−0.043	0.021	−0.086	−0.004
High (+1 SD)	−0.085	0.028	−0.146	−0.035
Low (−1 SD)	−0.012	0.026	−0.063	0.039
SEM as a moderator				
Index of moderated mediation	0.118	0.025	0.078	0.173
High (+1 SD)	0.058	0.028	0.010	0.118
Low (−1 SD)	−0.149	0.031	−0.217	−0.094

Note: Bootstrap sample size = 5000. Abbreviations: SDM, stress-is-debilitating mindset; SEM, stress-is-enhancing mindset.

## Data Availability

The data presented in this study are available on request from the corresponding author. The data are not publicly available due to privacy.

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
