# Peer review of "Promoting or Prohibiting? Investigating How Time Pressure Influences Innovative Behavior under Stress-Mindset Conditions"

_behavsci, 2024, doi:10.3390/bs14020143_

Round 1
Reviewer 1 Report
Comments and Suggestions for Authors
The paper titled " Promoting or Prohibiting? Investigating How Time Pressure Influences Innovative Behavior Under Stress Mindsets Conditions" appears intriguing. However, it does have some significant shortcomings that the authors should address. My comments are provided in numbered form below.
The introduction could benefit from a more direct approach to presenting the research problem. It currently takes a while before the main focus and significance of the study are made clear to the reader. There is a lack of a strong, clear thesis statement or hypothesis in the introduction. This makes it challenging for readers to understand the specific purpose and direction of the research from the outset.
the introduction includes relevant literature, it tends to list these studies without a critical analysis or a clear demonstration of how they directly relate to the current study’s objectives. The introduction could be improved by more explicitly outlining the gap in the current body of research that this study aims to address, thereby highlighting its unique contribution.
There is a lack of critical analysis in the literature review. While many sources are cited, there is minimal discussion on their limitations, contradictions, or the specific gaps they leave, which this research aims to address. The review could benefit from more explicit connections between the cited literature and the study's theoretical framework. Linking existing research more directly to the theoretical underpinnings of the paper would strengthen its foundation. The literature review seems to focus predominantly on established studies, with less emphasis on the most recent research. Including more current studies could provide a more updated view of the field and demonstrate the paper’s relevance in the context of ongoing research.
The methods section lacks a detailed description of the process for selecting the sample population. Greater transparency about the sampling methodology would enhance the credibility of the research findings. The section on data analysis is somewhat brief and could benefit from a more in-depth explanation of the analytical techniques used. A more detailed account of the statistical methods and their justification would provide clarity and robustness to the study.
The results section, while comprehensive in its statistical analysis, could benefit from a more narrative approach. This would help in interpreting and contextualizing the data, making it more accessible and understandable for readers not specialized in statistical analysis. the paper presents a range of statistical data, there is a lack of critical discussion about the implications of these findings. Expanding on how these results contribute to the existing body of knowledge or could be applied in practical settings would enhance the section's relevance.
Thank you!
Comments on the Quality of English LanguageSee the detail comments
Author Response
Dear reviewer:
Thank you for your valuable comments for our paper. Your comments and suggestions are quite helpful in improving the quality of this paper. We have revised the manuscript in accordance with your comments and suggestions.
Please take a look at the "Author's Notes File", which presents our point-by-point responses to the comments.
Thank you very much for your professionalism and patience.

Reviewer 2 Report
Comments and Suggestions for Authors
This is a well-written and thoughtful paper that touches on an important topic, especially after workforce disruptions from COVID. That said, I feel that the authors could have gone further in their exploration of the topic. What they are driving at is personality characteristics that make people more or less sensitive to stress and their interpretation of it. Arousal theory/arousal gradients can be considered here, in where people set their ideal arousal level, and how that gets adjusted over time (aging, experience, other factors in one’s life, etc.). This could have been better explored, both in the literature, and perhaps in the sample from which the data were collected. There are a number of studies now looking at executive functioning and mindfulness, and how that intersects with anxiety and creativity/innovation. As mindfulness is something that can worked on, and has been shown to decrease arousal and increase innovation, I think it would have been a good addition.
It is unfortunate that a more physiological representation of stress was not collected – such as cortisol. I do have some concern about the measures used regarding stress, happiness, and innovation. Regarding innovation measures in particular, I refer you to the article Measuring Innovative Work Behavior by de Jong and den Hartog (2010) in Creativity and Innovative Management. It is an older article, though newer than the 1994 measure used, that provides a good background of the measures at that time. More generally, I would have liked to understand in more detail why the particular measures were selected, and if they were translated into the local language, was the translation and tool validated ahead of its use.
The hypotheses were straightforward and well documented, though perhaps expected based on prior psychological literature. Regarding sample, it seemed to be a sample of convenience, though not specified. In addition, how were the participants recruited and informed of the study. Did they receive any benefit by participating; did they sign and informed consent? I was concerned that their participant number was directly linked to them through their phone number; hopefully this information remains in a secured place. All of this can have an impact on results generalizability. Further, it would be helpful if a power analysis was conducted and p values adjusted accordingly, given the fairly large sample size.
In reading through the results, I assume that 86.7% effective rate was the response rate (line 291)? How were missing data handled? Was there any difference between those who participated and those who did not (assume the companies had records providing percent breakdown of males/females, age, job tenure, etc). Information on line 292-297 could be placed into a table for easier viewing. Also, while some of the results are significant, and assuming the power analysis does not change that, I am unsure of the significance (for example, line 388, simple slope of -0.015. I think this needs to be taken into account in the discussion, especially when you talk about practical application. How useful is that difference?
Lastly, as mentioned above, the discussion and conclusion should be sensitive to the measures, results and other literature areas that you can bring in, extending the theory and linkage between physiological/psychological stress, personality traits, and innovation. Again, focusing on a practical way an company or person can help regulate that, like mindfulness, would benefit employers and employees.

Author Response

(The authors gave the same response as above.)

Reviewer 3 Report
Comments and Suggestions for Authors
After reviewing the report, I think the paper is enough valid about the hypothesis, results, and discussion. Probably from statistical point of view the number of the participants (390) is quite low. However, the statistical procedure is fine. I know is very difficult to increase the number of participants to assure the conclusion about the relationship between pressure and innovations behaviour. Therefore, from my point of view It should add in the conclusion and further discussion.
Author Response

(The authors gave the same response as above.)
